# Barriers and facilitators to utilizing HIV prevention and treatment services among migrant youth globally: A scoping review

Kevin Li[1,2,3], Natasha Thaweesee[1,4,5], Allison Kimmel[1,3], Emily Dorward[1], Anita Dam[1,5]*

1 Office of HIV/AIDS, United States Agency for International Development, Washington, District of Columbia, United States of America, 2 Department of International Health, Johns Hopkins Bloomberg School of Public Health, Baltimore, Maryland, United States of America, 3 STAR, Public Health Institute, Washington, District of Columbia, United States of America, 4 Department of Epidemiology, Colorado School of Public Health, Aurora, Colorado, United States of America, 5 GHTASC, Credence LLC, Washington, District of Columbia, United States of America

* andam@usaid.gov

**Data Availability Statement:** All relevant data are within the paper.

**Funding:** The author(s) received no specific funding for this work.

## Abstract

Both migrants and young people experience disproportionately high rates of HIV acquisition and poor access to HIV prevention and treatment services. To develop effective interventions and reach epidemic control, it is necessary to understand the barriers and facilitators to accessing HIV services among migrant youth. We conducted a scoping review to identify these factors for migrant youth ages 15–24, globally. We conducted a PRISMA-concordant scoping review using keyword searches in PUBMED and Web of Science for peer-reviewed primary literature published between January 2012 and October 2022. We included studies that investigated barriers and facilitators to accessing services for migrant youth participants. We used the Socio-Ecological Model as an analytical framework. The 20 studies meeting the inclusion criteria spanned 10 countries, of which 80% (n = 16) were low- and middle-income countries. Study methods included were quantitative (40%), qualitative (55%), and mixed methods (5%). Six studies included refugee youth (30%), 6 included migrant worker youth (30%), 3 included immigrant youth (15%), 2 included rural migrant youth (10%), and 1 included immigrants and refugees. The remainder represented unspecified migrant youth populations (10%). At the individual level, education level and fear of infection acted as barriers and facilitators to HIV services. At the relationship level, social support and power in relationships acted as barriers and facilitators to HIV services. At the community level, barriers to HIV services included discrimination and stigma, while community and religious outreach efforts facilitated access to HIV services. At the structural level, barriers to HIV services included stigmatizing social norms, lack of health insurance, and legal barriers. Migrant youth face significant, unique barriers to accessing HIV services. However, facilitators exist that can be leveraged to enable access. Future implementation science research, enabling policies, and adapted programmatic interventions should prioritize migrant youth as a distinctive sub-population to receive targeted HIV services.

**Competing interests:** The authors have declared that no competing interests exist.

## Introduction

Young people aged 15 to 24 contribute 29% (630,000) of all new HIV infections globally [1]. Young people are at an increased risk of HIV acquisition due to developmental, psychologic, social, and structural transitions [2, 3]. The burden of HIV among sub-populations of young people varies drastically across geographical locations, with adolescent girls and young women experiencing greater rates of HIV acquisition in sub-Saharan Africa, and young men who have sex with men at a greater risk of HIV acquisition in the United States, Europe, Latin and Central America, and the Asia and Pacific region [1, 3–6]. Disparities in HIV acquisition risk among young people can be attributed to low utilization of HIV services when compared to adult populations, with studies finding that youth populations have lower HIV testing rates, low antiretroviral therapy (ART) uptake and continuation rates, and high loss to follow-up for HIV services [7–9]. These outcomes result in poor viral load suppression and increased rates of infection [10]. Identified barriers to care among young people include stigma, negative attitudes from caregivers and healthcare providers, and unstable guardianship [7, 10, 11].

Migrants experience disproportionately high rates of HIV infection when compared to their non-migrant counterparts, with a global HIV prevalence ratio of 1.70, 2.37, 3.98, and 54.79 for foreign-born migrants, refugees, undocumented people, and asylum-seekers, respectively [12]. They are a priority population group for which targeted interventions are required for ending the HIV epidemic [13]. Migrants face numerous challenges in accessing HIV services across all levels of the care continuum, experiencing delayed initiation into HIV care, poor retention in HIV services, and frequent interruptions in treatment, which negatively impacts viral load suppression and increases the risk of developing HIV drug resistance [13–17]. Identified challenges to utilizing HIV care include language barriers, stigmatizing cultural norms, legal status, and discrimination by healthcare providers [18–21].

Similarly, these barriers have also been observed among both male and female migrants aged 15 to 24 [15]. Migrant youth face unique challenges related to migration as they adjust to a new environment during a period of life marked by transformative personal development. Their heightened vulnerability due to their age results in an increased risk of exploitation, violence, and unsafe employment when compared to non-migrant youth and older migrants [22–24]. Traumatic experiences associated with migration and a loss of support systems make migrant youth more susceptible to xenophobic and discriminatory treatment, leading to worse mental health outcomes [25–27]. There are also marked disparities in HIV risk between older migrants and migrant youth, with individuals who migrated as youth having a greater likelihood of acquiring HIV compared to individuals who migrated after the age of 36 [28].

Though evidence exists of adult migrants' barriers to, and facilitators of, accessing HIV services, less is evident about the unique factors that affect migrant youth access to HIV services. This review seeks to understand what is known about the barriers to, and facilitators of, accessing HIV prevention and treatment services among migrant youth to inform effective interventions.

## Methods

Due to the exploratory nature of our topic, we chose to conduct a scoping review for its ability to map the existing body of literature and to identify existing gaps in literature with greater flexibility than a systematic review. With the goal of better summarizing the diversity and depth of the lived experiences of migrant youth (defined as 15 to 24 years), a scoping review also allows for the inclusion of both quantitative and qualitative study designs. No review protocol has been registered. Our guidelines for conducting a scoping review were based upon the procedures outlined by previously conducted scoping reviews and were structured in

accordance with the Preferred Reporting Items for Systematic Reviews and Meta-Analyses (PRISMA) guidelines for scoping reviews (S1 Checklist) [29–31].

## Eligibility criteria

We included primary research of all study designs across all geographical regions. Studies were included based on the following inclusion criteria:

1. Published in a peer-reviewed journal

2. Written in English

3. Published between January 1, 2012 and September 30, 2022

4. Reports primary data collection and analysis

5. Includes migrant youth populations (defined as ages 15 to 24)

6. Discusses HIV prevention and treatment services

7. Discusses barriers or facilitators to HIV prevention and treatment services

Publications that had a study population not limited to the youth age range were included if at least 70% of the study participants were ages 15 to 24.

## Information sources and search

On October 19, 2022, we searched peer-reviewed literature from two databases (PubMed and Web of Science) published between January 1, 2012, to September 30, 2022. Our search strategy was built in three steps using free-text and controlled vocabulary (e.g. MeSH terms): 1) migration (e.g. "Transients and Migrants" OR "Refugees*" OR "Immigrants*") AND young people (e.g. "Adolescent" OR "Youth") AND HIV (e.g. "human immunodeficiency virus" OR "acquired immunodeficiency syndrome"). The search strategy was adapted for both databases (S1 Text).

## Selection of sources of evidence and data charting

We imported all studies into Covidence which automatically removed all duplicates, which was later verified during title and abstract screening [32]. Title and abstract screening was conducted by two reviewers (KL and NT) using the inclusion criteria listed above. Publications that were accepted by both reviewers were subjected to full-text review, while situations where the two reviewers disagreed on the inclusion of a publication were resolved by a third reviewer (AD or AK). Publications accepted by both reviewers then underwent data extraction. Using an 18-item template on Covidence, study characteristics (e.g., country of origin, study population, study design) and reported barriers and facilitators to accessing HIV prevention and treatment services were captured by one reviewer (S1 Table). After concurrence with a second reviewer was obtained, charted data was considered ready for analysis.

## Data items

Based on the charted data from the included publications, we used Microsoft Excel to analyze our findings using qualitative content analysis and matrix analysis, and synthesized our data using a narrative approach as outlined by Hsieh and Shannon [33, 34]. By extracting meaning units from the results section of the publications, we were able to develop codes to categorize the various barriers and facilitators to accessing HIV prevention and treatment services mentioned. Quotations from qualitative studies were used to exemplify the meaning of each

developed code when present. Codes were initially developed with an inductive approach, as the result sections were examined without any predetermined themes. Throughout the review process, a deductive approach was adopted to better utilize the Social-ecological Model (SEM) as an analytical framework [35]. Upon discussion with team members, meaning units were categorized and sorted by the levels of the SEM (e.g., individual, community, structural). When possible, barriers and facilitators were separated by the type of migrant population studied.

## Results

### Study characteristics

Fig 1 shows a PRISMA flowchart of the results of our initial search, title and abstract screening, and full-text review processes. We identified 1583 studies across PubMed and Web of Science, of which 72 were duplicates. The remaining 1511 studies underwent title and abstract screening, resulting in 282 studies assessed for eligibility through full-text review. After completing our full-text review, 20 publications were included for data extraction and synthesis [36–55].

Table 1 shows the characteristics of the 20 included publications, of which two used data from the same study. Ten studies were conducted with a qualitative study design, eight studies with a cross-sectional quantitative research design, and one study with a qualitative and cross-sectional mixed methods study design. Study settings included Uganda (six publications), China (four publications), Thailand and the United States (two publications each), Canada, Nepal, Nigeria, South Africa, Switzerland, and Vietnam (one publication each). Eighty percent of studies were conducted in an LMIC setting (16 publications), while 20% were conducted in high income countries (four publications).

A wide range of migrant populations were represented in the included publications, including six focusing on refugees, six on migrant workers, three on immigrants, two on rural-to-urban migrants, one on immigrants and refugees, and two publications that did not specify the type of migrant population. Twelve publications focused on both male and female migrant youth populations, seven were specific to female migrant youth, and one was specific to male migrant youth.

### Barriers and facilitators to accessing HIV services using the SEM

Fig 2 outlines the identified barriers and facilitator3s to accessing HIV services when mapped out to each level of the SEM. Of the 20 publications, 18 mentioned barriers and facilitators at the individual level (e.g., education, socioeconomic status), four mentioned factors at the social level (e.g., relationship dynamics, social support), 11 mentioned factors at the community level (e.g., outreach, service delivery), and six mentioned factors at the structural level (e.g., health insurance policy, legal precarity). Seventy percent of publications (n = 14) reported solely on barriers, while 30% (n = 6) identified both barriers and facilitators to engaging with HIV services.

### Individual

Twelve publications reported education as a factor which influences access to HIV services [38, 39, 42, 44, 45, 47, 49, 51–55]. We used the word "education" to encompass two types of knowledge: knowledge about HIV transmission and prevention, and general knowledge acquired through access to education, including language and literacy. A lack of knowledge about HIV and sexual and reproductive health was linked to decreased preventive behavior in nine of the 12 publications and included rural migrants, immigrants, migrant workers, and

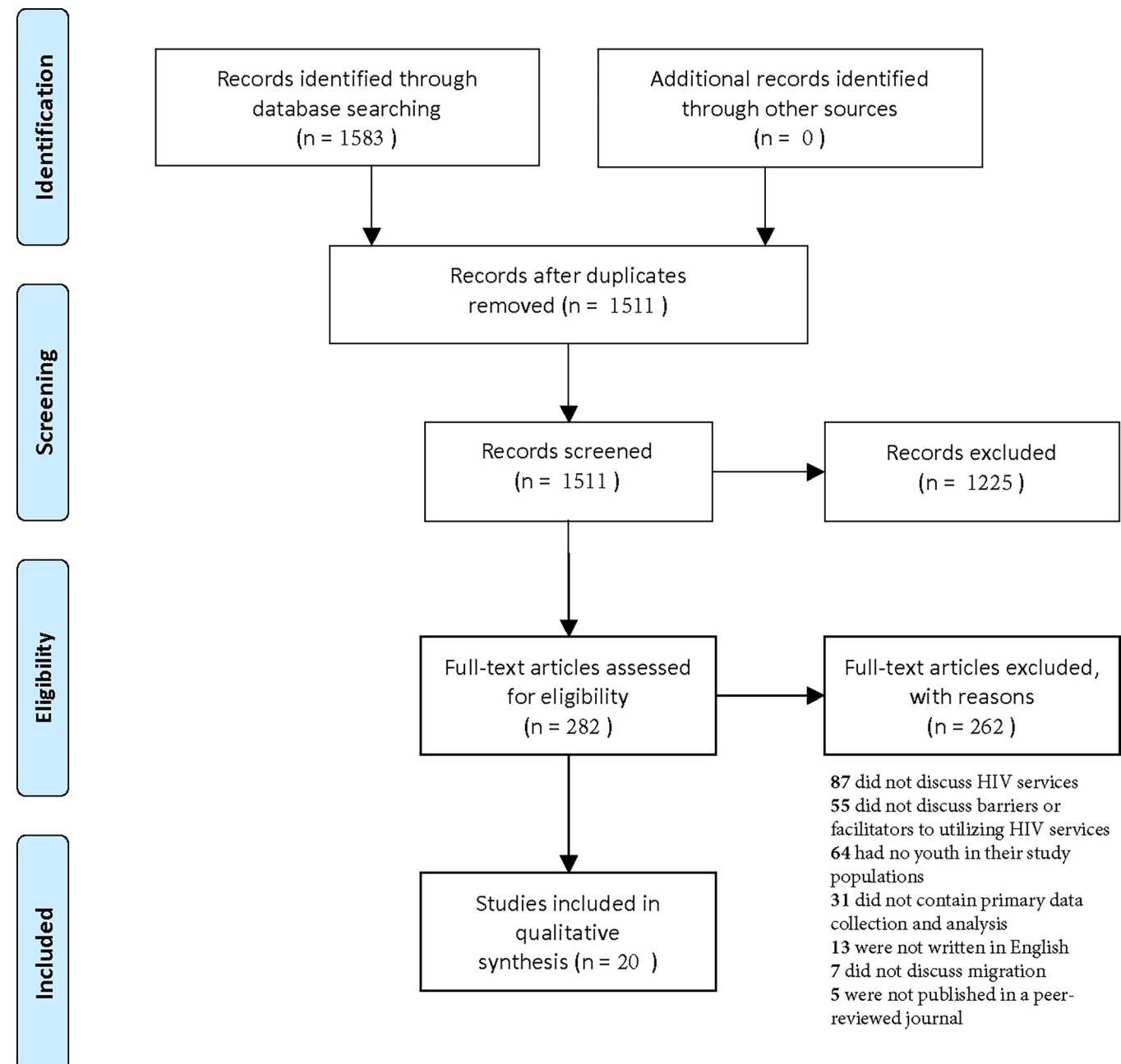

**Fig 1. PRISMA flow diagram of the scoping review process.**

refugee youth study populations [38, 39, 42, 45, 47, 51, 53–55]. For example, one publication described a reluctance among migrant sex worker youth in China to see a doctor to treat AIDS/STIs due to a lack of knowledge of healthcare [55]. Four studies cited poor access to standard education as a limiting factor to accessing HIV services [39, 47, 51, 55]. For example, a study conducted among refugee youth in Uganda identified poor literacy rates among the study population as a barrier to engaging with HIV testing and prevention services [47]. Conversely, a higher education level was associated with a greater willingness to engage with HIV services in four studies with immigrant and refugee study populations [44, 49, 52, 53]. A study

**Table 1. List of studies included in the scoping review, including authors and year, country of origin, study design, type of migrant population, age range, and sex.**

| Lead Author, year | Study Location | Study Design | Age Range in Years (15–24; ≥ 70% under 24) | Sex | Type of Migrant Population |
|---|---|---|---|---|---|
| Bam et al., 2013 [36] | Nepal | Qualitative | 15–45 | Male | Migrant workers |
| Bernays et al., 2020 [37] | South Africa | Qualitative | 16–24 | Both | Non-specified |
| Cai et al., 2013 [38] | China | Quantitative | 16–27 | Female | Rural migrants |
| Ha et al., 2022 [39] | Vietnam | Quantitative | 18–29 | Female | Migrant workers |
| Haderxhanaj et al., 2014 [40] | USA | Quantitative | 15–24 | Both | Immigrants |
| Huang et al., 2014 [41] | China | Quantitative | 18–29 | Female | Migrant workers |
| Huang et al., 2016 [42] | China | Quantitative | 16–24 | Female | Rural migrants |
| Ivanova et al., 2019 [43] | Uganda | Mixed method | 13–19 | Female | Refugees |
| Kingori et al., 2018 [44] | USA | Qualitative | 18–25 | Both | Immigrants and refugees |
| Kunnuji et al., 2013 [45] | Nigeria | Quantitative | 10–19 | Female | Non-specified |
| Logie et al., 2019 [46] | Uganda | Quantitative | 16–24 | Both | Refugees |
| Logie et al., 2021 [47] | Uganda | Qualitative | 16–24 | Both | Refugees |
| Logie et al., 2021 [48][a] | Uganda | Qualitative | 16–24 | Both | Refugees |
| Logie et al., 2022 [49] | Uganda | Quantitative | 16–24 | Both | Refugees |
| Logie et al., 2022 [50][a] | Uganda | Qualitative | 16–24 | Both | Refugees |
| Manoyos et al., 2016 [51] | Thailand | Qualitative | 16–24 | Both | Migrant workers |
| Mellini & Mileti, 2022 [52] | Switzerland | Qualitative | 18–25 | Both | Immigrants |
| Ngobi et al., 2020 [53] | Canada | Qualitative | 18–29 | Both | Immigrants |
| Tangmunkongvorakul et al., 2017 [54] | Thailand | Qualitative | 15–24 | Both | Migrant workers |
| Zhuang et al., 2013 [55] | China | Qualitative | 14–45 | Female | Migrant workers |

[a] These publications originated from the same study

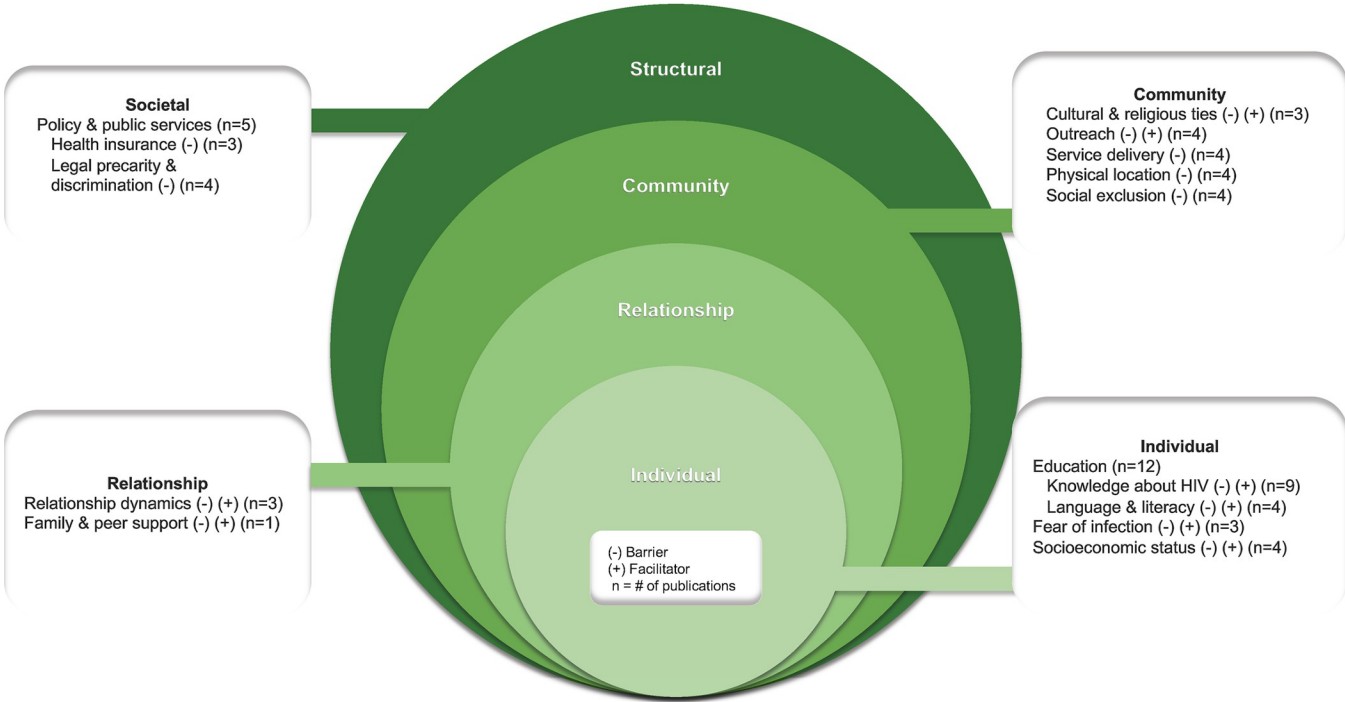

**Fig 2. Diagrammatic representation of barriers and facilitators to HIV service use by migrant youth and adolescents, organized by the SEM.**

conducted with second-generation sub-Saharan African immigrant youth in Switzerland found that higher education levels facilitated greater discussion about testing for HIV prior to discontinuing condom use with a sexual partner [52].

Three studies identified fear of infection as a factor which influences access to HIV services among immigrant, refugee, and unspecified migrant youth [37, 48, 53]. Fear of infection was influenced by the impact of an HIV diagnosis on the study population's identity and emotions. One study among young migrants in South Africa identified a reluctance to seek HIV services to avoid being marked by the stigma associated with a positive HIV status, finding their self-concept, described by Bernays et al. as "soon-to-be-successful" youth, to be in direct opposition with being at-risk of HIV acquisition [37]. Migrant youth in this study were found to adopt an optimistic and ambitious personal narrative in relation to their future accomplishments as a method of staying hopeful despite the challenging environment they inhabit. Thus, risking a positive HIV test result by engaging in testing services could decrease their optimistic outlook. Similarly, a study conducted with African immigrants in Canada found that the stigma associated with HIV caused a reluctance to seek HIV testing services for fear of testing positive [53]. Conversely, the same study found that migrant youth who did not express concerns about stigma were more likely to access HIV testing services.

Four studies reported socioeconomic status as a factor which influences access to HIV services among refugee, rural migrant, and migrant worker youth study populations [41, 42, 46, 49]. In all four studies, employment was linked to greater engagement and knowledge of HIV services. This link was shown to vary based on industry of employment and income. For example, a study of rural migrants in China found that migrant youths working in entertainment venues had greater self-efficacy for accessing HIV prevention services when compared to factory and restaurant workers, due to greater exposure to sexual health education through the industry's proximity to sex work [42]. However, a study among migrant sex worker youth in China found a disparity in access to public health services between youth working in affluent entertainment venues and those working in non-affluent entertainment venues, potentially due to a greater reach of public health programming among affluent venues [41].

**Relationship.** Three publications identified dynamics within a romantic relationship as a factor which influences access to HIV services among immigrant and refugee youth [47, 49, 52]. For example, in a study focused on refugee youth in Uganda, the likelihood of seeking HIV testing was dependent on the current strength of their romantic relationship, with a supportive intimate partner relationship encouraging testing to foster trust and security, and a precarious intimate partner relationship discouraging testing for its potential to end the relationship [49]. Another publication using data from the same study highlighted how decision-making power within a heterosexual relationship influenced HIV testing behavior, with women lacking agency in the relationship being reluctant to seek HIV testing for fear of negatively impacting their marriage [47]. However, the same study data found that women experiencing physical intimate partner violence (IPV) were three times more likely to have tested for HIV, perhaps due to an increased chance of receiving HIV testing while seeking medical care for IPV-related injuries. Engagement in HIV testing was also mentioned as a method of empowerment within a relationship, further facilitating future testing behavior.

One study on migrant youth in Uganda reported familial and peer support as a factor which influences access to HIV services, with fear of judgment from family members discouraging participants from seeking HIV testing [49]. This lack of support was further compounded by a reluctance to inform or engage family members in the decision-making process to accessing HIV services for fear of potential coercion. Support from trusted peer navigators, self-identified refugee youth with experience working in local communities as health or as

peer educators, was identified as a preferred avenue of social support and promoted test-taking behavior by avoiding the stigma associated with HIV.

**Community.**　　Three studies mentioned religious and cultural ties as a factor which influences access to HIV services among immigrants and refugees [44, 47, 52]. The relationship between the cultural values of the study participants' origin country and the country in which they currently reside was dependent on geography. In a study of refugee and immigrant Somali youth in the United States, Islamic religious norms were associated with increased stigma towards sexual health, reluctance to seek out sexual health education, and increased difficulty in acclimating to American cultural norms [44]. A study of sub-Saharan African immigrant youth in Switzerland found that second-generation immigrants were more comfortable negotiating condom use when compared to their parents [52]. In a study conducted with refugee youth in Uganda, religious beliefs were a simultaneous barrier and facilitator to accessing HIV services [47]. Upholding religious beliefs from their origin country allowed participants to retain ties to their community, serving as an avenue of social support and increased HIV education outreach. Conversely, conservative religious leaders posed a barrier to accessing HIV services by using their influence to discourage HIV prevention methods.

Four studies referred to community outreach as a factor which influences access to HIV services among immigrants and refugees [43, 44, 47, 53]. For example, in a study of refugee and immigrant Somali youth in the United States, collaboration with faith-based organizations and religious leaders could facilitate access to HIV services through educational outreach towards parents to reduce stigma [44]. Similarly, a study conducted with adolescent refugee girls in Uganda found that participants did not know where to seek care because of inadequate outreach efforts [43]. Outreach efforts led by untrusted figures were found to discourage access to HIV services among refugee youth in Uganda due to a reluctance to engage with the medical community and workers offering HIV testing services [47].

Four studies reported inadequate service delivery strategies as a barrier to accessing HIV services among refugees and migrant workers [36, 43, 47, 54]. HIV service facilities lacking privacy were cited as a barrier in two studies among immigrant female youth [43, 47]. For example, a study conducted with refugee adolescent girls in Uganda found that participants did not want to return to the facilities due to a lack of private spaces and a lack of trusted adults in whom they could confide [43]. Two studies conducted with migrant worker youth noted a lack of targeted and effective prevention programming, citing an absence of information material in their native language and a lack of condom availability [36, 54].

Four studies indicated the physical location of care facilities as a barrier to accessing HIV services among migrant workers and refugees [36, 39, 43, 47]. For example, a study conducted with young migrant workers in Vietnam indicated that there was limited access to HIV prevention services near the industrial zones in which they reside [39]. Similarly, Nepalese migrant workers in India cited an inability to access HIV services due to a lack of voluntary counseling and testing facilities near the country border [36]. Inaccessible transportation to HIV testing services was noted as a barrier to care for refugee youth in Uganda [47].

Five studies indicated social exclusion as a barrier to accessing HIV services among migrant workers, immigrants, and refugees [30, 40, 42, 47, 51]. Among Nepalese migrant workers in India, rejection due to caste-related stigma was cited as a reason for migration which reduced access to HIV services [30]. In a study conducted with refugee youth in Uganda, stigma related to sexual and reproductive health was only negatively associated with HIV testing among adolescent girls and young women and not adolescent boys and young men, indicating a gendered impact [42]. Another study among migrant sex workers in China, HIV-related stigma posed a barrier to seeking HIV testing and treatment services through gossip and social exclusion among co-workers while also increasing financial precarity through the risk of losing clients [55].

**Structural.**    Five studies listed policy and public services as barriers to accessing HIV services among immigrants, refugees, and migrant workers [40, 48, 51, 53, 54]. In a study conducted with Spanish-speaking immigrant youth in the United States, lack of health insurance coverage was noted as a prevalent barrier to accessing HIV services, correlating with low healthcare utilization when compared to non-Hispanic white youth [40]. For migrant workers in Thailand, a lack of healthcare insurance resulted in missed encounters for HIV prevention education in clinics [51]. Even for migrant worker youth in Thailand with health insurance, difficulty navigating a foreign healthcare system resulted in a lack of awareness of eligibility rights to access youth-friendly HIV-related health services [54].

In a study conducted with urban refugee youth in Uganda, legal precarity was listed as a barrier to accessing HIV services [48]. The criminalization of sex work and same-sex relationships resulted in a reluctance to seek care due to the risk of persecution, and a positive HIV diagnosis could result in the denial of a visa for those seeking asylum. Refugee youth also experienced discrimination by healthcare workers who expressed xenophobic sentiment towards them. For migrant worker youth in Thailand without work permits, fear of deportation resulted in a mistrust of public health educators due to suspicion that they were government officials [51, 54]. In a study conducted with African immigrants in Canada, fear of prosecution due to the criminalization of HIV non-disclosure resulted in a reluctance to seek HIV testing services [53].

## Discussion

The scoping approach of our review allowed for the mapping of barriers and facilitators to accessing HIV services among young migrants across multiple countries, encompassing different migrant subpopulations, and inclusive of multiple study designs. The review's use of the SEM to synthesize our findings allowed for a multi-level examination of barriers and facilitators to accessing HIV services and their interrelation.

### Barriers

To our knowledge, this scoping review is the first to map what is known about barriers and facilitators to use of HIV prevention and treatment services for migrant youth globally. This review highlights that the most frequently identified barriers to HIV service access were at the individual level of the SEM, with lack of education, either generally or about HIV, being the most commonly reported (60%). A systematic review of barriers and facilitators to HIV testing among migrants in high income countries, regardless of age, similarly found that lack of knowledge negatively impacted usage of HIV testing services [14]. The focus on the individual level identified in this review is consistent with literature pertaining to migrants regardless of age, with a systematic review of migrants living with HIV in Organization for Economic Co-operation and Development (OECD) countries finding that 64% of included publications reported barriers at the individual level impacting the HIV services cascade [56]. This trend for barriers at the individual level is unexplained but noted broadly in sexual health research [57]. A study analyzing 324 abstracts presenting observational research on adolescent sexual health behavior found that 95% of studies included individual barriers, with under 30% of studies presenting information on other levels of the SEM despite the importance of adopting a multi-level ecological approach [57]. This greater representation is also noted downstream at the intervention level, with a review examining 157 health education and behavior intervention articles finding that studies were more likely to report interventions at the individual level [58]. Despite this, both authors noted the importance of considering environmental and multi-level factors when understanding adolescent sexual risk behavior and developing effective

interventions, respectively. To develop equitable and effective public health solutions, future research should supplement this relative lack of emphasis on non-individual factors [57, 59].

Across all levels of the SEM, stigma was observed as a barrier to HIV services in 35% of publications (n = 7). Fear of infection, lack of provider training, and misconceptions of how HIV is transmitted have all been identified as drivers of stigma [60–62]. Similarly, our review identified fear of infection and poor knowledge of HIV as barriers to accessing care [38, 39, 42, 45, 47, 48, 51, 53–55]. Social norm enforcement exists as a major driver of stigma for migrant youth, where deviation from the norm results in an 'othering' from a community or social group, as evidenced in our review by the reporting of cultural and religious norms discouraging access to HIV services [44, 47, 63]. For young migrants seeking HIV prevention and treatment services, there is an intersection of stigmas between potential HIV infection status and migrant status. Stigma marking relates to barriers found at both the individual and structural level, with young migrants expressing reluctance to access healthcare due to the negative connotations associated with a positive HIV status and due to negative perceptions of migrants among local populations, respectively [37, 48].

Discriminatory and harmful policy (institutional stigma), alienation from the public, friends, and family (social stigma), and inequitable treatment by healthcare providers (professional stigma) create a hostile environment for the migrant youth that discourages utilization of HIV services [47–49, 52, 53, 55]. As a result, young migrants experience poor outcomes, including greater economic instability, legal precarity, and poor social protection [41, 48, 51, 53, 54].

While most barriers identified in this review do not appear to be unique to migrant youth, we noted the fear of an HIV diagnosis negating the optimism and ambition comprising the self-concept of young migrants in South Africa [37]. Additionally, age at migration appears to influence the acculturation process and educational outcomes, with younger migrants being more impacted by local cultural norms when developing their sense of identity and having a greater likelihood of achieving educational attainment [64, 65]. Presumably, this difference in experience and integration within local culture may impact the barriers and facilitators to care related to education and cultural and religious ties for young migrants. This difference may also have implications on the development of interventions tailored specifically to migrant youth that seek to improve educational outcomes, or that leverage the influence of these ties to advocate for greater engagement with HIV services.

## Facilitators

Facilitators to accessing HIV services were noted at the individual, relationship, and community levels, including higher education levels, social support, and targeted outreach efforts from community and religious organizations [44, 49, 50, 52, 53]. These findings can potentially be leveraged to mitigate equivalent barriers to accessing care (e.g., using peer support navigators to supplement the social support a young migrant may lack from their family or community due to stigma). No youth-specific facilitators were identified in this review that address inadequate service delivery or barriers at the structural level. Structural and service delivery-related facilitators among migrants living with HIV, regardless of age, include universal healthcare policy inclusive of documented and undocumented foreign-born individuals and a strong relationship between the patient and clinical care team [56]. In-depth interviews with key informants in California, including health care and social service providers, attorneys, and legal/policy experts, outline the success of medical-legal partnerships between HIV services clinics and legal teams at increasing the quality of HIV service delivery for immigrants by increasing a service provider's understanding and empathy towards the patient [66, 67]. Since

these publications did not disaggregate results by age, a future avenue for research may investigate the applicability of these facilitators and interventions for migrant youth populations.

The facilitators noted in this review were not exclusive to migrant youth, as greater education, social support, and the leveraging of community and religious leaders have been identified as facilitators to accessing HIV services for both young people overall and for migrants regardless of age [56, 68, 69]. While this may be due to a lack of research focusing on migrant compared to research focused on each population separately, the consistency of these findings may suggest that existing research and interventions on this topic for youth and migrants could be transferable to or adapted for migrant youth.

While the scope of our review included barriers and facilitators to HIV treatment services, young migrants living with HIV were not represented in our included publications. We speculate that this could be due to several factors, including a higher likelihood of later HIV presentation among migrants, as well as a potential underreporting of the number of migrant youth living with HIV due to poor engagement with testing services when compared to older migrants [70–74]. Because of this absence, future research should seek to identify if known barriers and facilitators to utilizing HIV treatment among older migrants are applicable to migrant youth. These barriers include stigma, lack of knowledge and language barriers, while facilitators include strong interpersonal relationships, perceived health improvement in relation to medication adherence, and strong patient-physician relationships [56, 75–77].

Differences in barriers and facilitators between study location related primarily to a country's cultural norms surrounding sexual health, with countries that normalize sexual health education facilitating access to HIV services among migrants. There were no other identifiable differences in barriers and facilitators related to geographic location, but certain areas were underrepresented in our included publications. Similarly, there were not enough publications examining migrant subpopulations within similar geographic or contextual environments to identify differences in barriers and facilitators to HIV services. Because of this gap in the literature, we cannot infer whether their similarities are due to a lack of data or to migrant subpopulations facing similar barriers and facilitators. Future research should continue to map out the experiences of the various subpopulations that fall under the 'migrant' umbrella across different economic, political, and geographic contexts to determine the need for tailored public health interventions.

## Strengths and limitations

While our objective was to conduct a global review, certain geographic locations were underrepresented or not represented in our included publications, including migration to and within countries in South America and Western European countries. Similarly, the diversity of lived experience captured under our study's definition of 'migrant' may not be fully represented by the migrant subpopulations found in our included publications. We thus have a highly heterogeneous study population, making it difficult to compare findings across multiple publications in different contexts. Since our objective was to identify gaps in the literature, these limitations reveal avenues for future research and areas that should receive greater focus to provide a more comprehensive analysis.

The scope of our review may have excluded pertinent information about barriers and facilitators to accessing HIV services among young migrants. Since we chose to only search PubMed and Web of Science, we may have excluded relevant articles found in other databases. Our review only included studies that were published in English, which may have excluded relevant findings. Thirteen publications were excluded at the full-text review stage for not being written in English. Grey literature can report results that would not be found in peer-reviewed journals, including research with null or insignificant findings [78]. Because our review did

not include grey literature, it may be subject to publication bias, wherein there is an overreporting of positive results [79]. Nonetheless, our review identified common barriers and facilitators to HIV services across various migrant youth subpopulations and geographic contexts, indicating a level of internal consistency in our findings.

## Conclusion

Migrant youth are disproportionately at risk of HIV infection and subsequent poor health outcomes due to poor access to HIV prevention and treatment services. Our results identify what is currently known about multilevel barriers and facilitators to utilizing HIV services. While migrant youth face many similar barriers to care as both young people and migrants separately, they also face unique challenges attributed to the intersection of these two identities, particularly in relation to acculturation and identity formation. Gaps in literature and a limited number of studies focusing on migrant youth highlight a need for future research to identify facilitators to care that can be leveraged to develop effective interventions. To achieve the 95-95-95 testing and treatment targets set by UNAIDS for all subpopulations and age groups, further emphasis should be placed on the development of policies and interventions that address both the individual and structural factors that influence access to HIV prevention and treatment services [80].

## Supporting information

**S1 Checklist. PRISMA-ScR checklist.**
(DOCX)

**S1 Text. Search term keywords.**
(DOCX)

**S1 Table. Data extraction template.**
(DOCX)

## Acknowledgments

We would like to thank Teresa Yeh for her input on the development of our search strategy. We would like to thank Dr. Caitlin Kennedy, Dr. Mary Latka, Dr. Benny Kottiri, and Dr. George Siberry for their input on and review of our manuscript. The views and opinions in this analysis are those of the authors and do not necessarily represent the views of USAID, PEPFAR, or the United States Government.

## Author Contributions

**Conceptualization:** Natasha Thaweesee, Allison Kimmel, Emily Dorward, Anita Dam.

**Data curation:** Kevin Li, Natasha Thaweesee.

**Formal analysis:** Kevin Li, Natasha Thaweesee, Allison Kimmel, Anita Dam.

**Investigation:** Natasha Thaweesee.

**Methodology:** Kevin Li, Natasha Thaweesee.

**Project administration:** Kevin Li.

**Software:** Kevin Li.

**Supervision:** Anita Dam.

**Writing – original draft:** Kevin Li, Natasha Thaweesee, Anita Dam.

**Writing – review & editing:** Kevin Li, Natasha Thaweesee, Allison Kimmel, Emily Dorward, Anita Dam.

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
