## [Decision Letter · Decision Letter 0]

10 Oct 2023

PGPH-D-23-01205

Barriers and facilitators to utilizing HIV prevention and treatment services among migrant youth globally: a scoping review

Dear Dr. Dam,

Thank you for submitting your manuscript to PLOS Global Public Health. After careful consideration, we feel that it has merit but does not fully meet PLOS Global Public Health’s publication criteria as it currently stands. Therefore, we invite you to submit a revised version of the manuscript that addresses the points raised during the review process.

We look forward to receiving your revised manuscript.

Kind regards,

Kevin Escandón, MD, MSc

Academic Editor

Journal Requirements:

1. Please provide separate figure files in .tif or .eps format only and remove any figures embedded in your manuscript file. Please also ensure all files are under our size limit of 10MB.

 Editor Comments:

This review has merit and overall sound methodology. However, it does need substantial rewriting and better description of the methods. A higher level of detail is expected for this type of review.

I think all reviewers have reasonably criticized methods aspects so please make sure to address them entirely one by one and adjust the search as needed. Please make sure to transcribe the reviewer comments from a marked-up sanitized pdf attached into your itemized response letter.

Reviewers' comments:

Reviewer's Responses to Questions

**Comments to the Author**

1. Does this manuscript meet PLOS Global Public Health’s publication criteria? Is the manuscript technically sound, and do the data support the conclusions? The manuscript must describe methodologically and ethically rigorous research with conclusions that are appropriately drawn based on the data presented.

Reviewer #1: Yes

Reviewer #2: Partly

Reviewer #3: Partly

2. Has the statistical analysis been performed appropriately and rigorously?

Reviewer #1: N/A

Reviewer #2: N/A

Reviewer #3: N/A

3. Have the authors made all data underlying the findings in their manuscript fully available (please refer to the Data Availability Statement at the start of the manuscript PDF file)?

Reviewer #1: Yes

Reviewer #2: No

Reviewer #3: Yes

4. Is the manuscript presented in an intelligible fashion and written in standard English?

Reviewer #1: Yes

Reviewer #2: Yes

Reviewer #3: Yes

5. Review Comments to the Author

Reviewer #1: The topic of the study is relevant, the intersection of young people and immigrants is a global public health problem that needs to be tackled to mitigate the HIV epidemic. The manuscript is well written and its methodology is sound.

Major comment:

Search strategy - Although I understand the difficulties of reading articles in languages other than English, given that most massive immigration events take place in non-English-speaking countries (“global South”), I think that the decisin of excluding articles in other languages in the Search ("language filter") imposes an important limitation on the work. The authors could have carried out the search without excluding articles on the basis of language and excluding them in a second step. By doing that, readers could at least have an idea of the amount of information/articles that were not evaluated on the basis of language. This strategy would also be better suited to a global perspective and would show the intention to be more inclusive of other countries and communities.

Minor comments

Methods: Add reference to Covidence software.

Results – Figure 2 – Suggest adding sample sizes (n) in the boxes indicating the number of studies reporting each Variable

Limitation paragraph (lines 501-503): “Thirteen publications were excluded at the full-text review stage for not being written in English, but 85% of included publications (n=17) were published from a study location that does not use English as an official language.” I was confused, the language was not used as a filter in the Search? If yes, those 13 publications had “scaped” the filter? Moreover, it is hard to conclude anything meaningfull based on those on the statement that 85% of the publications included were from non-English-speaking countries.... the real question is how many publications were not included due to the language restriction. Even more, although the purpose and the methods of this study are sound, it would be valid to raise some discussion on how we do not have data on locations where forced and massive immigration is hapenning.

Reviewer #2: Did the authors collaborate with a public health or medical librarian to formulate the search terms?

What was the rationale behind excluding articles published before 2021?

When was the search strategy put into action?

Is there a specific reason for utilizing only two databases, which might have potentially resulted in overlooking significant publications?

For studies where the participant age range extended beyond youth, how was the 70% threshold determined for the 15-24 age group?

The inclusion criterion, "Discusses HIV prevention and treatment services," lacks context.

The same ambiguity applies to the subsequent inclusion criterion, "Discusses barriers or facilitators to HIV prevention and treatment services."

Reviewer #3: Thank you for the opportunity to read this scoping review on this important topic. Understanding the experiences of migrant youth in preventive HIV/STI behaviours is important. I have provided comments throughout the manuscript PDF which has been uploaded for your review.

6. PLOS authors have the option to publish the peer review history of their article (what does this mean?). If published, this will include your full peer review and any attached files.

**Do you want your identity to be public for this peer review?** For information about this choice, including consent withdrawal, please see our Privacy Policy.

Reviewer #1: No

Reviewer #2: No

Reviewer #3: No

---

## [Editor Report · Decision Letter 1]

5 Jan 2024

Barriers and facilitators to utilizing HIV prevention and treatment services among migrant youth globally: a scoping review

PGPH-D-23-01205R1

Dear Ms. Dam,

We are pleased to inform you that your manuscript 'Barriers and facilitators to utilizing HIV prevention and treatment services among migrant youth globally: a scoping review' has been provisionally accepted for publication in PLOS Global Public Health.

Best regards,

Kevin Escandón, MD, MSc

Academic Editor